# Children in the household and risk of severe COVID-19 during the first three waves of the pandemic: a prospective registry-based cohort study of 1.5 million Swedish men

Agnes af Geijerstam [1], Kirsten Mehlig,[1] Monica Hunsberger,[1] Maria Åberg,[1,2] Lauren Lissner[1]

¹School of Public Health and Community Medicine, University of Gothenburg Institute of Medicine, Goteborg, Sweden
²Regionhälsan, Region Västra Götaland, Gothenburg, Sweden

**Correspondence to**
Dr Agnes af Geijerstam;
agnes.af.geijerstam@gu.se

## ABSTRACT

**Objective** To investigate whether Swedish men living with children had elevated risk for severe COVID-19 or infection with SARS-CoV-2 during the first three waves of the pandemic.

**Design** Prospective registry-based cohort study.

**Participants** 1 557 061 Swedish men undergoing military conscription between 1968 and 2005 at a mean age of 18.3 (SD 0.73) years.

**Main outcome measures** Infection with SARS-CoV-2 and hospitalisation due to COVID-19 from March 2020 to September 2021.

**Results** There was a protective association between preschool children at home and hospitalisation due to COVID-19 during the first and third waves compared with only older or no children at all, with ORs (95% CIs) 0.63 (0.46 to 0.88) and 0.75 (0.68 to 0.94) respectively. No association was observed for living with children 6–12 years old, but for 13–17 years old, the risk increased. Age in 2020 did not explain these associations. Further adjustment for socioeconomic and health factors did not attenuate the results. Exposure to preschool children also had a protective association with testing positive with SARS-CoV-2, with or without hospitalisation, OR=0.91 (95% CI 0.89 to 0.93), while living with children of other ages was associated with increased odds of infection.

**Conclusions** Cohabiting with preschool children was associated with reduced risk for severe COVID-19. Living with school-age children between 6 and 12 years had no association with severe COVID-19, but sharing the household with teenagers and young adults was associated with elevated risk. Our results are of special interest since preschools and compulsory schools (age 6–15 years) in Sweden did not close in 2020.

## BACKGROUND

Early evidence showed that children were less affected by the SARS-CoV-2 virus than adults and adolescents.[1 2] This notwithstanding, with the initial attempts to curb transmission by school closures and lockdowns, children around the world saw their lives upended.

## STRENGTHS AND LIMITATIONS OF THIS STUDY

⇒ As schools remained open in Sweden during the COVID-19 pandemic, the effects of living with children or not constitute an important comparison.
⇒ Our large study population, based on validated registry data and including key covariates reflecting the conscript's comorbidities and physical condition, is an important strength.
⇒ The major limitation is that only men were included.
⇒ Due to the observational design, we are also unable to rule out that unknown factors influenced the results.

The effects of school closures on the spread of infectious disease had been discussed before the COVID-19 pandemic. Modelling studies focusing on influenza showed the effectiveness varied depending on the basic reproduction number and on whether children were driving the attack rates due to less immunity compared with adults.[3–5] Despite the conflicting evidence on the effectiveness of school closures in relation to the character of SARS-CoV-2, closures were widely implemented as a non-pharmaceutical intervention (NPI).

Sweden was an exception where compulsory schools (ages 6–15 years) and preschools were kept open. Attendance was mandatory and enforced for compulsory school ages. Schools and preschools were to implement preventive measures such as distancing and hand hygiene and avoid unnecessary mixing of classes and teachers,[6] while teaching in upper secondary school was moved online.

Two large studies have been published on the risk of parental infection posed by living with children. Wood *et al* with a population of over 300 000 healthcare workers and their

families in Scotland,[7] and Forbes *et al* with the COVID-19: OpenSAFELY cohort of 12 million adults in England.[8] Neither could show an increased risk related to living with children of any age during wave 1. During wave 2, there was a small absolute risk associated with children of any age living at home, except for younger children as reported in the Scottish cohort. Part of the risk increase was attributed to the return to schools and preschools in September 2020.

Men are disproportionally affected by COVID-19, comprising 74% of those admitted to intensive care in Sweden.[9] In the present study, we had the opportunity to examine a large part of the male population in Sweden, for whom information is available on early health factors that influence severity of COVID-19.[10 11] As schools and preschools were open in Sweden, it is a unique starting point for investigating the effects of sharing a household with children during the pandemic.

## METHODS
### Study design
This is a prospective cohort study based on data from the Swedish Military Conscription Registry, combined with a socioeconomic population registry (LISA) from Statistics Sweden as well as the Swedish national hospital and intensive care registries.

### Population studied
The Swedish military conscript registry contains information about 1 949 891 Swedish individuals who enlisted for military service between late 1968 and 2005. During those years, Swedish law required all male citizens to enlist, except for those in prison or those with severe chronic somatic or psychiatric conditions or functional disabilities (approximately 2%–3% annually).

### Patient and public involvement
As this is a registry-based study, there has been no patient or public involvement.

## MAIN INDEPENDENT VARIABLES
### Children at home
Data about children registered at the same address as the men in the cohort was retrieved from Statistics Sweden. Most of the men are assumed to be fathers, biological or not, but they could also be grandfathers or lodgers. The age brackets correspond to Swedish preschool, primary school, middle school and lower and upper secondary school. No children in the youngest bracket would have started school during 2020–2021, but those aged 6 years in 2020 would have been in preschool during wave 1 and then in first grade during waves 2 and 3. School in Sweden is compulsory from 6 to 15 years of age and circa 85% of all 16–18 year-olds attend upper secondary school. Around 90% of all Swedish 2-year-olds attend preschool, with even higher rates for 4–5 year-olds.[12]

### Confounding variables
Weight and height were measured by standard anthropometric measurement techniques, and continuous body mass index (BMI) values ($kg/m^2$) were calculated, as BMI has been shown to be one of the major risk factors for severe COVID-19. Earlier studies on the same cohort have shown an association between BMI and cardiorespiratory fitness (CRF) in early adulthood and later risk of severe disease.[11 13–15] There is also a known correlation between CRF, height and BMI at conscription and the probability of having children.[16]

### Morbidity at baseline
All medical diagnoses prior to conscription are recorded in the conscript registry. Illness that could have affected the ability or decision to have children, as well as later risk of severe COVID-19 was controlled for using International Classification of Diseases (ICD) codes for respiratory disease, cardiovascular disease, diabetes, kidney disease and malignant cancers.[16 17]

### Socioeconomic indicators
Parental education was considered a proxy for socioeconomic position of the household. Using data from the LISA registry, it was based on the highest educational achievment in the household and categorised as: low (up to 9 years), medium (upper secondary school diploma with ≤2 years at university) and high education (≥3 years at university). Data on home municipality were collapsed into three categories: large, medium and small towns and municipalities.[18] Disposable family income was categorised into low, medium and high income based on tertiles.

From LISA, we also had information on profession. High-risk occupations were defined a posteriori as those where the risk of hospitalisation due to COVID-19 was 0.28% and higher. These included healthcare personnel, bus drivers, restaurant workers, service personnel, industrial workers, social workers and primary school teachers (n=314 834).

### Analytic sample
Originally comprising 1 949 891 conscripts, the cohort was reduced to 1 559 187 men after exclusions (figure 1). A total of 1 557 061 had information in LISA on children at home. Those who died during 2020 and until February 2021 (n=5012) were censored in the analysis prior to each wave, giving an at-risk population of 1 555 835 at the beginning of wave 1, 1 552 040 at wave 2 and 1 549 514 at wave 3. Characteristics of the population, together with crude outcome data, are shown in table 1.

## OUTCOME VARIABLES
### Hospitalisation due to COVID-19
Using the Swedish personal identification number, the full sample was linked to the National Patient Register and the Swedish Intensive Care Registry. From these, all cases between March 2020 and September 2021 with a

af Geijerstam A, *et al. BMJ Open* 2022;**12**:e063640. doi:10.1136/bmjopen-2022-063640

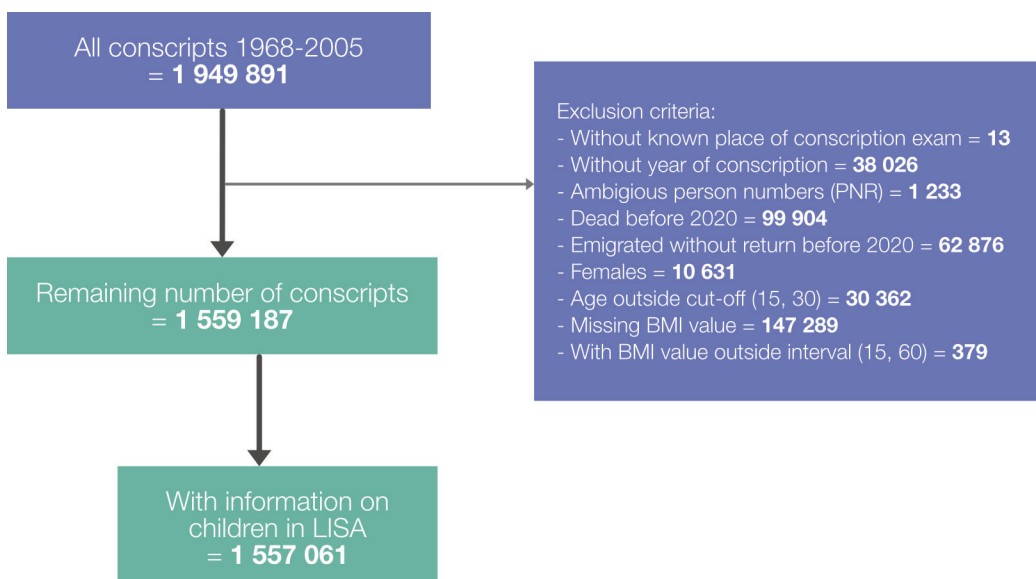

**Figure 1** Creation of analytical sample. BMI, body mass index.

main diagnosis of ICD U071 for test verified infection with SARS-CoV-2 and U072 for clinically diagnosed COVID-19 were identified. Records with U071 or U072 as a secondary diagnosis were counted as cases if the main diagnosis was clinically related to COVID-19 (online supplemental table S1). All illness requiring hospital care was considered severe COVID-19. Register data based on Swedish hospital records have high validity.[19]

### Infection with COVID-19

Free PCR testing began in summer 2020. Previously, testing was mainly done in hospitals. Therefore, earlier data are limited and not representative of the actual infection rates, as is seen in the comparison between testing and hospitalisations in figure 2. All positive tests were to be registered in the Sminet registry according to the Swedish Communicable Diseases Act. Data from

**Table 1** Characteristics of the study population

| Conscription year | 1968–1975 | 1976–1985 | 1986–1995 | 1996–2005 | All decades |
|---|---|---|---|---|---|
| N in 2020 | 282 828 | 440 991 | 495 775 | 337 467 | 1 557 061 |
| Age at conscription, mean (SD) | 18.5 (0.65) | 18.3 (0.82) | 18.3 (0.76) | 18.2 (0.58) | 18.3 (0.73) |
| Age in 2020 mean (SD) | 65.9 (1.94) | 57.5 (2.98) | 48.0 (3.00) | 38.3 (2.91) | 51.8 (9.89) |
| **Children at home (age in 2018)** | | | | | |
| Child of any age | 43 220 (15.3) | 177 630 (40.3) | 333 582 (67.3) | 212 025 (62.8) | 790 604 (50.8) |
| 0–3 years (%) | 292 (0.1) | 3 151 (0.7) | 38 873 (7.9) | 111 926 (33.2) | 154 242 (9.9) |
| 4–6 years (%) | 457 (0.2) | 5 657 (1.3) | 62 168 (12.5) | 92 332 (27.4) | 160 612 (10.3) |
| 7–10 years (%) | 1 275 (0.5) | 16 576 (3.8) | 124 800 (25.2) | 79 418 (23.5) | 222 069 (14.3) |
| 11–15 years (%) | 3 967 (1.4) | 47 911 (10.9) | 168 507 (34) | 35 318 (10.5) | 255 703 (16.4) |
| 16–17 years (%) | 3 597 (1.3) | 35 946 (8.2) | 68 234 (13.8) | 5 549 (1.6) | 113 326 (7.3) |
| 18–20 years (%) | 4 719 (1.7) | 40 733 (9.2) | 50 320 (10.2) | 2 641 (0.8) | 98 413 (6.3) |
| 20 and older (%) | 34 259 (12.1) | 94 456 (21.4) | 51 782 (10.4) | 15 983 (4.7) | 196 480 (12.6) |
| **Hospitalisations due to COVID-19** | | | | | |
| N, March 2020–September 2021 (%) | 2 261 (0.80) | 3 179 (0.72) | 2 455 (0.50) | 760 (0.23) | 8 655 (0.56) |
| With children at home (%) | 426 (18.8) | 1 318 (41.5) | 1 616 (65.8) | 463 (61) | 3 823 (44.2) |
| **PCR-confirmed infection with SARS-CoV-2** | | | | | |
| N, March 2020–September 2021(%) | 18 471 (6.5) | 50 272 (11.4) | 72 878 (14.7) | 47 825 (14.2) | 189 446 (12.2) |
| With children at home (%) | 3 793 (20.5) | 24 279 (48.3) | 54 920 (75.4) | 32 671 (68.3) | 115 663 (61.1) |

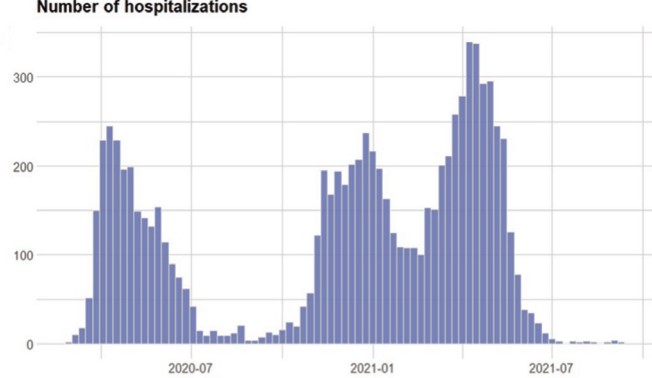

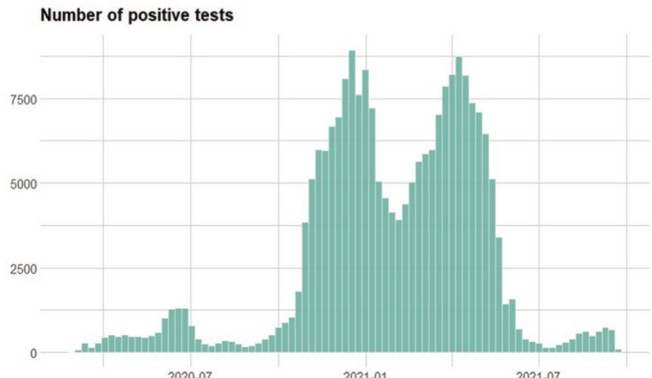

**Figure 2** Weekly admissions to hospital due to COVID-19 in Sweden and weekly registered PCR-verified infections between March 2020 and September 2020. Note that tests were not widely available until July 2020.

Sminet were extracted on 21 September 2021 and covers all positives until that date.

## Statistical analysis
The main independent variable was analysed as a binary variable in each age interval. Logistic regression was used to calculate the odds for hospitalisation and infection due to COVID-19 by this exposure category, adjusted for children in other age brackets, place of residence, income and profession in 2018, linear, quadratic and cubic terms of BMI, as well as CRF, height, parental education and chronic disease at conscription. All regression models were adjusted for exact age at conscription and year of conscription examination, and thus indirectly for age in 2020. Because overall events of hospitalisation were rare, we used penalised likelihood estimation (Firth method) to reduce (potential) small sample bias in maximum likelihood estimation

Statistical analyses were performed with SAS V.9.4 (SAS Institute). Statistical significance was set at 0.05 (two-sided tests).

## RESULTS
### Hospitalisations
Having a child of preschool age at home had a protective association with hospitalisation due to COVID-19 during the pandemic from March 2020 to July 2021 (figure 3). This association is statistically significant in the first and third wave with OR=0.63 (95% CI 0.46 to 0.88) and 0.75 (95% CI 0.60 to 0.94). During the second wave, no association could be seen (OR=1.00 (95% CI 0.75 to 1.33)). In contrast, no associations between living with children of primary school age and hospitalisation were observed overall or during any wave. Sharing the household with children 13 years of age or older conveyed an overall excess risk of hospitalisation, particularly during the second wave.

None of the results was attenuated after adjustment for covariates (online supplemental table S2). Exposures describing children of different age groups were not mutually exclusive, as many fathers live with more than one child. For the combined waves, the protective association for hospitalisation due to COVID-19 was strengthened to OR 0.54 (95% CI 0.42 to 0.71) when we considered a separate category of households with only preschool-aged children (online supplemental table S3).

### Infection
Living in a household with children over 5 years was associated with an increased risk for infection with SARS-CoV-2. The same pattern can be seen for all waves, with ORs significantly higher for all age groups apart from the preschool children who had a significant protective association (figure 4) with OR=0.91 (95% CI 0.89 to 0.93). Wave-specific results can be found in online supplemental table S4.

Adjustment for high-risk profession in the main analysis did not attenuate the effect estimates. To evaluate the effects of isolating with children not yet in compulsory school, we examined the subset of men in high-risk occupations only: having younger children still gave a protective association with OR=0.94 (95% CI 0.91 to 0.98). Again, when looking at those living with preschool children only, the association was stronger with OR=0.73 (95% CI 0.69 to 0.77).

## DISCUSSION
In this paper we demonstrate a robust association between residing with children of preschool age and a lower risk of severe COVID-19 during two of the three waves of the pandemic in Sweden.

The pattern for older children varied, but no significant associations were seen at any time among those aged 6–12 years. Exposure to the age group 13–17 years was associated with a higher risk of severe COVID-19 during the second and third waves. Sharing a household with a teenager or young adult was associated with a higher risk of disease during all waves.

There were significantly higher odds for testing positive for SARS-CoV-2 associated with living with children 6 years and older, except for the age group 6–8 years during the second wave. The age group 13–17 years presents the largest risk for infection and includes both children who

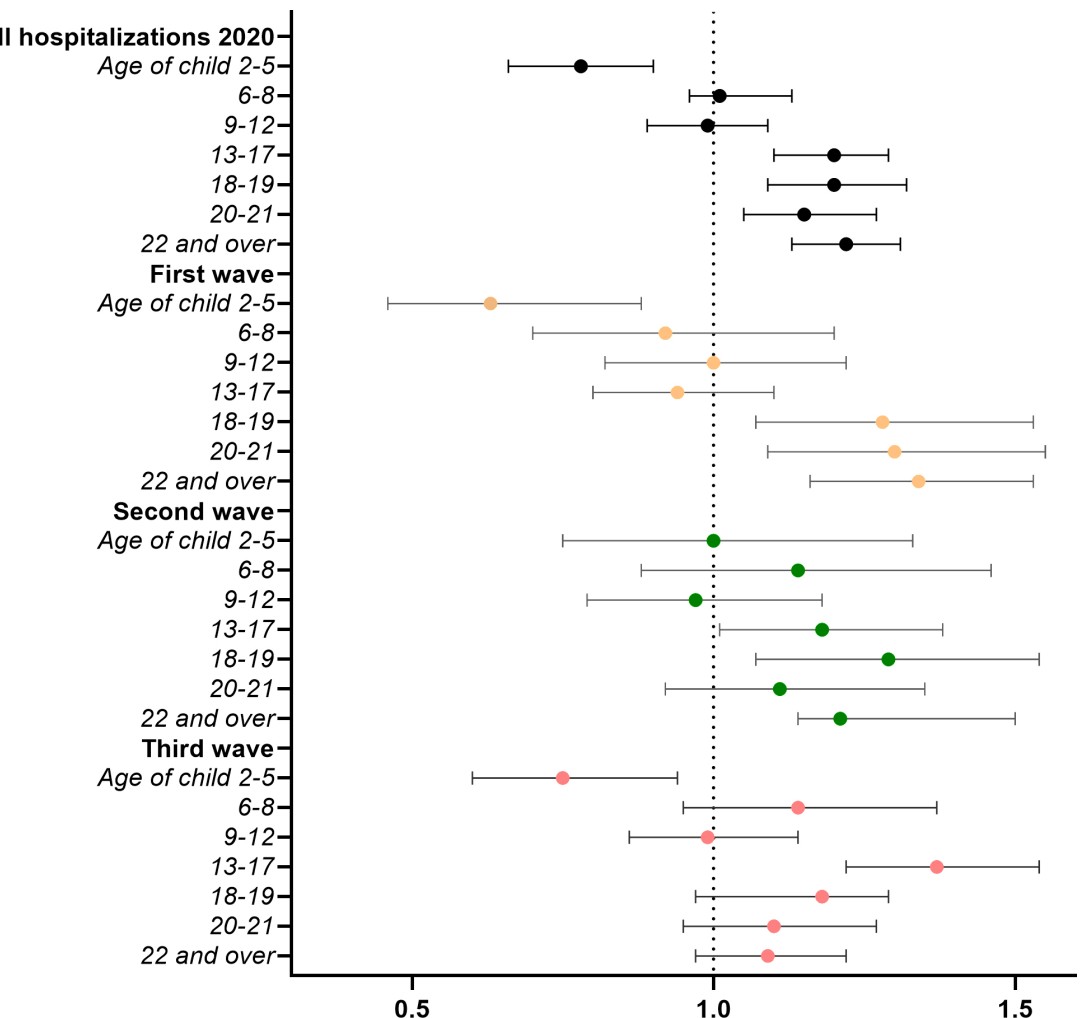

**Figure 3** Associations between children in the household and hospitalisation due to COVID-19 (n=1 557 061). ORs with 95% CI. Model controlled for children in other age groups, age, baseline BMI, CRF, height, chronic morbidity, parental education, income, profession and place of residence in 2018. BMI, body mass index; CRF, cardiorespiratory fitness.

attended school (age 13–15 years) and who had distance learning (age 16–17 years). It has been shown that adolescents transmit COVID-19 disease similarly to adults in households, and the combination of slightly older children in open schools could explain this pattern.[7 20–26] Unfortunately, the testing data do not include negative

results; therefore, it is not possible to analyse whether the infection rates partly mirror an increased testing frequency in certain groups.

The comparable studies from Scotland and England showed that having children at home (any age in England, 0–11 in Scotland) was not associated with increased risk

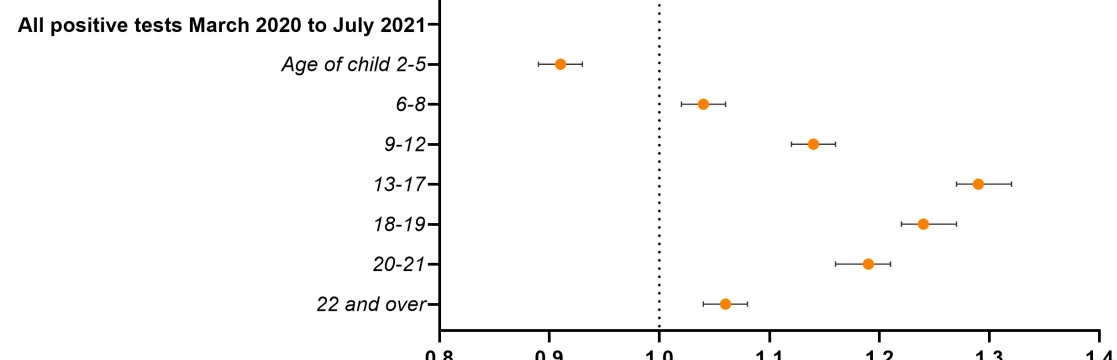

**Figure 4** Associations between children in the household and testing positive for SARS-CoV-2 March 2020–September 2021 (n=1 557 061). ORs with 95% CI. Model controlled for conscript's age, BMI, CRF, height, chronic morbidity, parental education and place of residence in 2018. BMI, body mass index; CRF, cardiorespiratory fitness.

of infection or severe disease in spring 2020. During the second wave risk of infection and COVID-19 related hospital admission was increased in the OpenSAFELY cohort but not in the Scottish cohort.[7 8] This finding was attributed in part to the school closures during spring and the reopening after summer. Our results fit the same pattern but cannot be explained the same way since the schools were open. An earlier Swedish study presents a similar small increase in infections among parents of lower secondary school age children during the first wave.[27]

High-risk profession did not affect the associations. With our smaller age brackets, we were able to single out associations in parents of children in preschool, which were distinctly different from those with children of older ages and highlights a methodological strength of our study.

No national figures of school or preschool attendance are available, but in the numbers of the official statistics bureau of Gothenburg (pop. 583 056), school attendance was distinctly lower in March 2020, and sick leave was higher during the pandemic compared with 2019. The possibility to self-isolate with younger children, together with more social distancing and better use of other protective measures in these families could of course contribute to the protective association, particularly considering that the effect was more pronounced in those with no older children.

Preschool, in contrast to school attendence is not mandatory, and the Swedish preschool teacher's union reported a large drop in attendance in March 2020. After March though, the majority returned,[28] making it unlikely that a large proportion of parents were isolating with their children. The fact that few of the preschool fathers also lived with children older than 13 years does limit their exposure to infection due to transmission in school, which also could contribute to the protective association. When the model is not adjusted for presence of older children, the association is stronger with OR 0.85 (95% CI 0.83 to 0.87), but it is still significant in the main model where the presence or lack of older children is accounted for.

Our large study population, based on validated registry data and including key covariates reflecting the conscript's comorbidities and physical condition, is an important strength. However, since this is an observational study, we cannot rule out that unknown factors such as behavioural differences between those living with or without children of different ages influenced the results. The major limitation is that only men were included in this study. Furthermore, all information from the LISA registry is from 2018. However, only minor changes in these covariates can be expected during 2019–2020, except for the oldest children who might have left home during this period.

We were not able to document the potential impact of vaccinations, as the main part of our cohort were not vaccinated until May 2021. Thus any vaccination effects would be restricted to the end of wave 3, with very few cases overall.

## Clinical and public health implications

The decision to keep mandatory schools open in Sweden offers a rather unique opportunity to compare our findings with observations in other settings. A recent review of studies trying to evaluate the effect of school closures on community transmission concluded that 'The true independent effect of school closures from the first wave around the world may simply be unknowable'.[29] The model calculations included in the review all had problems differentiating between NPI's implemented simultaneously. Closing schools was meant to control community transmission by limiting transmission between children and subsequently between children and parents. As our study shows, the OR for infection was higher for men living with children of all age groups except those aged 2–5 years. This could be expected due to the greater number of contacts in school but is still comparable with transmission patterns where schools were closed during spring 2020.

The finding that living with preschool children was associated with lower risk of hospitalisation due to COVID-19 does raise questions. If the effect is not entirely due to behavioural differences or parental health, it could be speculated that simultaneous infection with other respiratory viruses more commonly occurring in this group compared with older children,[30] such as rhinovirus, could be protective, as has been shown in vitro.[31 32] Both wave 1 and wave 3 coincide with the months when the Swedish Social Insurance Agency normally distributes the most compensation for care of sick child (mainly due to cold viruses in early spring).

## CONCLUSION

Young children seem to have played a minor part in the community transmission of COVID-19, even though preschools remained open in Sweden. As this study shows, adult men living in the same household as children of this age group had a lower risk both of infection and severe sickness. Having children between 6 and 12 years in the household was associated with a small increase in odds of infection, but not with severe disease. Having teenagers in the household was associated with increased rates of infection as well as severe disease in their fathers. These associations are similar in magnitude to those reported in other settings where schools were closed.

**Contributors** AG, LL and MÅ initiated the project. AG and KM performed all statistical analyses. AG had main responsibility for writing the article. LL, MÅ, KM and MH contributed to the structure and content of the manuscript and have read and approved of the final draft. LL and MÅ share last authorship. AG acts as guarantor.

**Funding** This work was supported by the EpiLife-Teens Research Program (FORMAS-2012-00038), by the Swedish ALF-agreement (ALFGBG-720201 and ALFGBG-813511), the Swedish Research Council (02508, VRREG 2019-00193,

2020-05792) and Swedish Research Council for Health, Working Life and Welfare (2021-00304).

**Disclaimer** The funding sources had no role in study design, collection, analysis and interpretation of data, the writing of the report or the decision to submit the article for publication. AG affirms that the manuscript is an honest, accurate and transparent account of the study being reported and that no important aspects of the study have been omitted.

**Competing interests** None declared.

**Patient and public involvement** Patients and/or the public were not involved in the design, or conduct, or reporting, or dissemination plans of this research.

**Patient consent for publication** Not applicable.

**Ethics approval** The study conforms to the principles outlined in the Declaration of Helsinki. The Ethics Committee of the University of Gothenburg and Confidentiality Clearance at Statistics Sweden approved the study (EPN Reference numbers EPN 462–14 and 567–15; T174-15, T653-17, T196-17, T2019-05875, T 2020–01325, T2020-02420, T2021-00797, T2021-03310). The requirement for informed consent was waived by the Ethics Committee of the University of Gothenburg for secondary analysis of existing data. The data were pseudonymised before being accessed by the study authors.

**Provenance and peer review** Not commissioned; externally peer reviewed.

**Data availability statement** Data may be obtained from a third party and are not publicly available. The data used in this study is available on request from The Swedish Defence Conscription and Assessment Agency and the National Board of Health and Welfare.

**ORCID iD**
Agnes af Geijerstam http://orcid.org/0000-0002-0897-6548

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
