## [Reviewer comments · BMJ Open]

This paper was submitted to a another journal from BMJ but declined for publication following peer review. The authors addressed the reviewers' comments and submitted the revised paper to BMJ Open. The paper was subsequently accepted for publication at BMJ Open.

(This paper received two reviews from its previous journal and two reviewers agreed to published their review.)

ARTICLE DETAILS

TITLE (PROVISIONAL)	Children in the household and risk of severe COVID-19 during the first three waves of the pandemic: a prospective registry-based cohort study of 1.5 million Swedish men.
AUTHORS	af Geijerstam, Agnes; Mehlig, Kirsten; Hunsberger, Monica; Åberg, Maria; Lissner, Lauren

VERSION 1 – REVIEW

REVIEWER	David A McAllister Wellcome Trust Intermediate Fellow, Professor of Clinical Epidemiology, University of Glasgow
REVIEW RETURNED	01-Feb-2022

GENERAL COMMENTS	This paper examined associations between the household composition of men – in terms of the number of children of different ages with whom they reside – and the risk of hospitalisation with COVID-19. This is an important topic because in a number of countries and regions restrictions remain in place concerning children and schools. Strengths of the study include:- - Sweden is an ideal setting to study this topic as schools remained open throughout- This is a nationally representative sample- The sample was large, allowing examining of risk by finely banded age strata- Excellent data on potential confounders, including BMI and occupation, which is missing from previous analyses Specific comments (not in order of importance) follow. 1. There is insufficient space in the introduction to discuss the pros and cons of the Swedish approach to pandemic management. This would be better addressed in an opinion piece. It is sufficient (and indeed important) for the interpretation of the findings, to note that schools (for children aged 6-15 years old) remained open and that attendance was compulsory. Incidentally, it would be useful to note whether compulsory attendance was enforced.2. Are there any data available on the number of vulnerable individuals sharing a household with children?3. Severe COVID-19 does not appear to be explicitly defined – it appears to refer to hospitalisation with COVID-19 but I did not find
--

	this explicitly stated in the text. 4. It is clearly explained in the methods section that each age-band was coded separately as a binary variable (1 for yes 0 for no, wherein an individual in a household with no children has zero for all of these variables). However, this is not obvious in the abstract. A sentence on this in the abstract and as a footnote to the tables would be helpful. 5. How did the analysis handle clustering of men within households? 6. The assertion that the effect on COVID-19 hospitalisation with having a pre-school child at home differ by season should be formally tested. 7. It would be useful for a virologist and/or immunologist to comment on the section on possible mechanisms for the observed associations. 8. A logistic regression model where any individual who died from another cause is excluded from the analysis correctly estimates neither the marginal probability nor the hazard function. One could use any method which appropriately accommodates time to event data such as Poisson regression (splitting the data into appropriate person-time intervals). 9. Age of the household members must be a time-varying covariate and given the narrow bands relative to the observation time a large % of children must move between the bands. I wonder if modelling this exposure as a time-varying covariate would be appropriate. 10. There is some colloquial English for example “calls are made” for “decisions are made”, “co-residing” for “living with” 11. The description of the findings of the Forbes et al and Woods et al (my own) paper are quite loose. In the case of the latter, we did not find an association between risk of COVID-19 infection and sharing a household with children aged 0-11 in August-October 2020, the hazard ratio was 1.03 and the 95% CI (95% CI 0.92 to 1.14). It would be better to directly quote effect estimates from these two studies along with a sufficient description of the analysis to allow the reader to compare and contrast the findings. 12. The conclusions may need to be revisited if the analysis is changed. However, even assuming they do not, I would be inclined to make a more modest claim in the conclusion such as “The observed increases in the risk of hospitalisation with COVID-19 associated with sharing a household with children of various ages were at most modest. Moreover, these associations were similar in magnitude to those reported in other settings where schools were closed. This finding does not favour the closure of schools as a method to reduce hospitalisation with COVID-19 among adults living with children.”
--	---

REVIEWER	Laurie Tomlinson, Associate Professor, LSHTM
REVIEW RETURNED	21-Feb-2022

GENERAL COMMENTS	This is an interesting and well-conducted analysis that builds on the strength of a prospective cohort with detailed information about baseline health status and further data such as occupation. I appreciate the way the authors present serially adjusted data enabling the reader to see how little impact various adjustments have on the crude analysis. The findings are largely consistent with studies from the UK although this study shows a lower risk of COVID associated hospitalisation for men living with pre-school aged children. In keeping with similar studies it suffers from limited availability of test data in the first wave before July 2020 making infection analyses potentially biased during this period, and limits understanding of who is previously unexposed
---

in subsequent waves. Due to slightly historic data being used there is also the risk of misclassification of whether people live with children or not.

Key to comparisons between parents and non-parents is that parents are healthier. Despite the cohort baseline data there is still uncertainty about the relative health status in this analysis during 2020-21. Firstly there is a long and variable lag between baseline measures of fitness and the study period so it is not clear how these track through life. I could not see the data presented – for example, what was the range of BMI in the conscripts? Were they all very fit at baseline? The authors adjust for ‘conscript’ height, BMI (via linear, quadratic, and cubic terms) and cardiorespiratory fitness and yet it is not clear how we know that these are associated with later life health above current measures of health status (e.g. are they diabetic?). COVID era morbidity is adjusted for using ICD-10 codes in broad categories but it is not clear how this data was captured. If from ICD-10 codes, was it only in people admitted to hospital? This would create misclassification by disease severity. It would be extremely helpful to see a table by parents/non-parents including the cohort measures of health and the ICD-10 parameters to compare markers of health status across the groups.

While the baseline cohort is large the overall mean age is 52 and the fact that this is a relatively lower risk group is reflected in the fact that there are only 8,541 hospitalisations – so the study is somewhat underpowered and does not look at ITU admission or death.

I am interested in the decision to exclude those who died of other causes which uses future time information and may create bias. Presumably those people could have been infected/admitted with COVID before dying of something else. If, as we would anticipate, these people are differentially represented in the non-parent group this would result in bias. Using non-COVID death as a control analysis would have provided an informative comparison of health status between the groups.

In relation to the finding that those living with the youngest children are less likely to be infected/hospitalised with COVID, these men are likely to be younger and age is far and away the strongest risk factor for COVID hospitalisation. Although the authors adjust for age it is not clear how they do so, or how good the age adjustment is given the relatively limited age range of the cohort.

In relation to generalisability, the cohort is of course only men, and of limited age range. Since people in poor health (2-3%) were not included in conscription the findings may not apply to people with longstanding health conditions who may be at greater risk from severe outcomes.

If available it would be helpful to present data about prevalence of infection within different age groups from community samples if available (like ONS-CIS or REACT in the UK) at different time periods of the pandemic to contextualise the risk within different age groups - how did adult rates compare to children?

I am not certain that the sensitivity analysis conducted among those with high-risk professions confirms either that the association is not affected by SES (Surely doctors, dentists, teachers etc must make up a fair chunk?) or that it shows that the association is not due to lower overall social contact among those with young children – there is no granular data on work patterns and men with very young children could be on parental leave or work part time – less risk of exposure even within high risk groups.

In the discussion the language could be considered causal. For example, “we demonstrate a robust protective association between co-residing with children of preschool age and the risk of severe

	COVID-19 during two of the three waves of the pandemic in Sweden” compared to “there is a lower risk of severe COVID-19 among men who live with children of pre-school age during two of the three waves of the pandemic in Sweden” I think that conclusions re the relationship to possible effects of other infections, and to policies around school closure are overstated. Showing similar OR across different age groups of children compared to rate ratios in countries with broader school closures does not show that school closures did not have an impact. A number of factors would need to be considered including absolute rates of infection, prior probability of undiagnosed infection in wave 1 and overall mitigation measures in schools. Indeed in England in the school closure in early 2021 about half of children were in ‘key-worker’ school. Similarly with regard to protective effects of other infections it is likely that this would be strongest in the 6-12 year old age group. If the null observed effect is due to it being offset by greater risk of infection then the health policy impact of this is unclear. Overall I think this is a good paper and the concerns above could be addressed. However, ultimately a lower risk among fathers of preschool age children could be plausibly explained by residual confounding from younger age, better (unmeasured) health status and fewer social contacts. Conversely the greater risk to parents living with older age children is consistent with the global literature. Of interest, I am not sure if the authors had seen this paper? https://www.medrxiv.org/content/10.1101/2021.02.28.21250921v1
--	--

VERSION 1 – AUTHOR RESPONSE

Answers to reviewer 1.

1. There is insufficient space in the introduction to discuss the pros and cons of the Swedish approach to pandemic management. This would be better addressed in an opinion piece. It is sufficient (and indeed important) for the interpretation of the findings, to note that schools (for children aged 6-15 years old) remained open and that attendance was compulsory. Incidentally, it would be useful to note whether compulsory attendance was enforced.

This is now edited in the text.

2. Are there any data available on the number of vulnerable individuals sharing a household with children?

Unfortunately, we don't have data on vulnerable individuals sharing a household with children.

3. Severe COVID-19 does not appear to be explicitly defined – it appears to refer to hospitalisation with COVID-19 but I did not find this explicitly stated in the text.

Thank you, this is now added to the text. Severe COVID-19 was defined as disease requiring hospitalisation or intensive care, see page 4.

4. It is clearly explained in the methods section that each age-band was coded separately as a binary variable (1 for yes 0 for no, wherein an individual in a household with no children has zero for all of these variables). However, this is not obvious in the abstract. A sentence on this in the abstract and as a footnote to the tables would be helpful.

It is now stated in the abstract that the reference was children of other ages or no children at all. We have also added a comment to the tables.

5. How did the analysis handle clustering of men within households?

We do not have information on whether several former conscripts live in the same household and cannot adjust for mutual correlation. However, it can be assumed that this situation occurs very rarely and is not likely to have an impact on the results.

6. The assertion that the effect on COVID-19 hospitalisation with having a pre-school child at home differ by season should be formally tested

This statement was a way to interpret the different results from the different waves, as the first and third wave to a large extent coincided with "cold season". This discussion is minimized in the revised version, due to word count limits.

7. It would be useful for a virologist and/or immunologist to comment on the section on possible mechanisms for the observed associations.

Thank you, we have reduced this section due to word limits, and mainly discuss possible mechanisms already mentioned in the references.

8. A logistic regression model where any individual who died from another cause is excluded from the analysis correctly estimates neither the marginal probability nor the hazard function. One could use any method which appropriately accommodates time to event data such as Poisson regression (splitting the data into appropriate person-time intervals).

We agree that the exclusion of men who died from causes other than COVID-19 is a potential source of bias, and we will comment on this in the text. For this reason, we redid the wave-specific analyses and included only men alive at the beginning of each time-interval, excluding all men who died before wave 2 in the analysis of wave 2 etc. This marginally changed the estimates.

9. Age of the household members must be a time-varying covariate and given the narrow bands relative to the observation time a large % of children must move between the bands. I wonder if modelling this exposure as a time-varying covariate would be appropriate.

We have emphasised the school category, rather than age of the child, given that no children in the youngest bracket would have moved from pre-school to primary school during the first three waves. School starts in late August the year a child turns 6. Approximately 20% of the children in the 13-17 bracket would have attended in-school teaching during spring 2020, and then distance learning, but as this group was already mixed, we chose to keep the model rather simple.

10. There is some colloquial English for example "calls are made" for "decisions are made", "co-residing" for "living with"

Thank you for pointing this out, we have removed colloquial English as suggested.

11. The description of the findings of the Forbes et al and Woods et al (my own) paper are quite loose. In the case of the latter, we did not find an association between risk of COVID-19 infection and sharing a household with children aged 0-11 in August-October 2020, the hazard ratio was 1.03 and the 95% CI (95% CI 0.92 to 1.14). It would be better to directly quote effect estimates from these two studies along with a sufficient description of the analysis to allow the reader to compare and contrast the findings.

Thank you for this, we have made changes in the text.

12. The conclusions may need to be revisited if the analysis is changed. However, even assuming they do not, I would be inclined to make a more modest claim in the conclusion such as "The observed increases in the risk of hospitalisation with COVID-19 associated with sharing a household with children of various ages were at most modest. Moreover, these associations were similar in magnitude to those reported in other settings where schools were closed. This finding does not favour

the closure of schools as a method to reduce hospitalisation with COVID-19 among adults living with children.”

This is now rephrased. Thank you for the suggestion.

Answers to reviewer 2

1. Key to comparisons between parents and non-parents is that parents are healthier. Despite the cohort baseline data there is still uncertainty about the relative health status in this analysis during 2020-21. Firstly there is a long and variable lag between baseline measures of fitness and the study period so it is not clear how these track through life.

I could not see the data presented – for example, what was the range of BMI in the conscripts? Were they all very fit at baseline?

The authors adjust for ‘conscript’ height, BMI (via linear, quadratic, and cubic terms) and cardiorespiratory fitness and yet it is not clear how we know that these are associated with later life health above current measures of health status (e.g. are they diabetic?).

Thank you for pointing this out. The paper now includes two references concerning early life health status and risk of severe COVID-19. Robertson et al(1) showed that BMI at conscription is strongly associated with later risk. In our earlier paper(2) we showed a protective association between higher cardiorespiratory fitness in late adolescence and later risk of severe COVID. Using later health testing data on the same individuals we also saw that fitness tracks over time.

As the conscription exam was mandatory for all Swedish men up until 2005, the ranges of BMI and fitness are representative of the male population. We will include a sentence in the text about this. The protection against severe disease might be explained by parents being healthier. But still, the rationale for closing schools was mainly the exposure to parents and community transmission. If the Swedish parents experienced a higher exposure due to their children being in school, we believe this would show in the testing and hospitalization data compared with the Forbes et al. and the Wood et al studies. Instead, our results largely mirror these studies.

2. COVID era morbidity is adjusted for using ICD-10 codes in broad categories but it is not clear how this data was captured. If from ICD-10 codes, was it only in people admitted to hospital? This would create misclassification by disease severity. It would be extremely helpful to see a table by parents/non-parents including the cohort measures of health and the ICD-10 parameters to compare markers of health status across the groups.

We controlled for chronic morbidity at baseline, i.e. at conscription. All ICD-codes for previous or current disease are present in the registry. This has been made clearer in the text.

3. While the baseline cohort is large the overall mean age is 52 and the fact that this is a relatively lower risk group is reflected in the fact that there are only 8,541 hospitalisations – so the study is somewhat underpowered and does not look at ITU admission or death.

Yes, our cohort was at lower risk than one including those older than 70. This is particularly seen in the death rates, which were very low. And more so among those with children at home since they are even younger.

4. I am interested in the decision to exclude those who died of other causes which uses future time information and may create bias. Presumably those people could have been infected/admitted with COVID before dying of something else. If, as we would anticipate, these people are differentially represented in the non-parent group this would result in bias. Using non-COVID death as a control analysis would have provided an informative comparison of health status between the groups.

Please see answer to reviewer 1, question 8.

5. In relation to the finding that those living with the youngest children are less likely to be infected/hospitalised with COVID, these men are likely to be younger and age is far and away the

strongest risk factor for COVID hospitalisation. Although the authors adjust for age it is not clear how they do so, or how good the age adjustment is given the relatively limited age range of the cohort. Age at baseline (conscripted) has little variation, so including a linear term is sufficient. What is important is age in 2020, and this varies because of year of baseline exam varies. We tested a model with age² in 2020, without any major changes in results.

6. In relation to generalisability, the cohort is of course only men, and of limited age range. Since people in poor health (2-3%) were not included in conscription the findings may not apply to people with longstanding health conditions who may be at greater risk from severe outcomes.

Thank you, the fact that individuals with long standing health problems were excluded from conscription is added to the limitations.

7. If available it would be helpful to present data about prevalence of infection within different age groups from community samples if available (like ONS-CIS or REACT in the UK) at different time periods of the pandemic to contextualise the risk within different age groups - how did adult rates compare to children?

Unfortunately, no community sampling of that type was conducted in Sweden.

8. I am not certain that the sensitivity analysis conducted among those with high-risk professions confirms either that the association is not affected by SES (Surely doctors, dentists, teachers etc must make up a fair chunk?) or that it shows that the association is not due to lower overall social contact among those with young children – there is no granular data on work patterns and men with very young children could be on parental leave or work part time – less risk of exposure even within high risk groups.

Thank you for this comment, it was unclear in the text and is now changed. The separate analysis of high-risk professions was meant to address the assumption that the protective association was only due to increased isolation among those who could work from home and keep their small children with them. Most children attended pre-school during 2020/21, after the initial drop in attendance in March 2020, which we have clarified.

9. In the discussion the language could be considered causal. For example, “we demonstrate a robust protective association between co-residing with children of preschool age and the risk of severe COVID-19 during two of the three waves of the pandemic in Sweden” compared to “there is a lower risk of severe COVID-19 among men who live with children of pre-school age during two of the three waves of the pandemic in Sweden”

Thank you for pointing this out, we have rephrased this.

10. I think that conclusions re the relationship to possible effects of other infections, and to policies around school closure are overstated. Showing similar OR across different age groups of children compared to rate ratios in countries with broader school closures does not show that school closures did not have an impact. A number of factors would need to be considered including absolute rates of infection, prior probability of undiagnosed infection in wave 1 and overall mitigation measures in schools. Indeed in England in the school closure in early 2021 about half of children were in ‘key-worker’ school. Similarly with regard to protective effects of other infections it is likely that this would be strongest in the 6-12 year old age group. If the null observed effect is due to it being offset by greater risk of infection then the health policy impact of this is unclear.

Thank you for these constructive objections. This discussion has been shortened in the revised manuscript and the conclusions made more modest.

11. Overall I think this is a good paper and the concerns above could be addressed. However, ultimately a lower risk among fathers of preschool age children could be plausibly explained by residual confounding from younger age, better (unmeasured) health status and fewer social contacts.

Conversely the greater risk to parents living with older age children is consistent with the global literature.

Thank you for this, we have made changes to the text. Also, see our answer to question 1.

VERSION 2 – REVIEW

REVIEWER	Hernandez-Suarez, Carlos M Universidad de Colima, Facultad de Ciencias
REVIEW RETURNED	12-May-2022

GENERAL COMMENTS	CHILDREN IN THE HOUSEHOLD AND RISK OF SEVERE COVID-19: A PROSPECTIVE STUDY OF 1.5 MILLION SWEDISH MEN. COMMENTS First at all, I want to congratulate the authors of this paper. I think you are on the right track. I want to declare to the authors that I have work extensively in transmission of infectious diseases in schools and other gathering places, and that I have contributed to the risk of infection of SARS-CoV-2 by age, gender, and other comorbidities. Having said that, I want to state that I believe that your conclusions are not well established, discussed enough. A secondary explanation is not mentioned well enough, and I believe that it explains your results. I don't think that you must accept it, only that I believe the discussion is not complete. The first sentence in the Results section in the abstract, mention "a protective effect", and I think that there are other possibilities that are not discussed enough. The authors suggest that a lower risk of infection with SARS-CoV-2 in adults is associated with the presence of pre-school children (p. 2, l. 55) in a household. They also suggest that living in a household with children over 5 was associated with an increased risk for infection (p.8, l. 38). Both assertions do not oppose and assume for a moment that they are true, then, the authors take the first one and suggest some sort of protection induced by children (p.10 ls. 10-18). But it is a well-known fact that young people (ages 14-18) have a higher contact rate, since they have a broader social life and interaction than children 2-5, and it is a wellknown fact that risk of infection is smaller in children less than 5 , thus, it well may be that the observed reduced risk of infection in adults is not due to the presence of 2-5 year old children in the household, but to the lack of older children. I urge the authors to consider and discuss this. I don't see a problem with the statistical analysis nor other issue of statistical nature, except the interpretation of results. Suggesting a protective effect may mislead other researchers to look in the wrong place. I don't think other explanations (as living with individuals with higher
---

	exposure) are discussed deeper in the paper, suggesting as the main explanation a protective effect of a biological nature. Good job and good luck.
REVIEWER	McAllister, David University of Glasgow
REVIEW RETURNED	18-May-2022
GENERAL COMMENTS	Having reviewed the changes since the BMJ submission and the point by point review I have no further concerns about the manuscript.

VERSION 2 – AUTHOR RESPONSE

To answer reviewer 1, we now included a more detailed discussion about the possible explanations for the protective effects seen between the presence of pre-school children and severe COVID-19 outcomes, as well as a positive test result. This includes your main concern that the reduction in risk of infection might not be due to the presence of pre-school children but rather to the absence of older children. In a model unadjusted for older children the protective association with pre-school children is given by $OR=0.85$ (95% CI 0.83-0.87). To investigate whether the protective association with young children was explained by the lack of older children in the family, conferring an increased risk for hospitalization, we present results from a mutually adjusted model including categories of children with all ages (ref = no children at home) (p. 8, 1.3 and 28). The protective association with young children was still observed albeit with reduced effect size compared to the unadjusted result, $OR=0.91$ (95% CI 0.89-0.93). Possible explanations for this finding are further discussed in the text (page 9).

VERSION 3 – REVIEW

REVIEWER	Hernandez-Suarez, Carlos M Universidad de Colima, Facultad de Ciencias
REVIEW RETURNED	08-Jul-2022
GENERAL COMMENTS	The main concern that I had in my first review, was related to attributing the findings to a "protective effect", without discussing the possibility that living with younger children was tantamount to living with a group with low social interaction. This part is now discussed. Still, there are two statements that I think should be reviewed. I provide an alternative phrasing: Page 9, row 3. it says: "In this paper we demonstrate a robust protective association between residing with children of pre-school age and the risk of severe COVID-19 during two of the three waves of the pandemic in Sweden. " I believe it should say something in the following lines:

	"In this paper we demonstrate a robust association between residing with children of pre-school age and a lower risk of severe COVID-19 during two of the three waves of the pandemic in Sweden." (I believe that the word "protective" is misleading) Also, in page 10, row 56, it says: "As this study shows, this age group conveyed a protective association both with the risk of infection and with the risk of severe sickness in adult men living in the same household." I believe it should say something in the following lines: "As this study shows, adult men living in the same household with children of this age group, had both a lower risk of infection and a lower risk of severe sickness." (Again, I believe that the word "protective" is misleading). Excellent work.
--	---

VERSION 3 – AUTHOR RESPONSE

To answer reviewer 1 we have omitted the word “protective” and rephrased the mentioned paragraphs accordingly. Thank you for pointing this out.